# OpenReview forum: "On the Unreasonable Effectiveness of Feature Propagation in Learning on Graphs with Missing Node Features"
_logconference.io/LOG/2022/Conference — LoG 2022 Poster_

### Official Review · Reviewer_DWdq · 2022-10-17

**Overall Score:** 8
**Confidence:** 4

**Review:**

The authors study the problem of graph representation learning with missing node features. They propose a simple yet effective method (Feature Propagation) to impute the missing features by iteratively diffusing the known features in the graph. They provide the theoretical justification to consolidate the Feature Propagation method. They conduct extensive experiments with a surprisingly high rate of missing features to validate the effectiveness of the proposed method.

Overall speaking, I find that the idea of this paper is interesting and the proposed method itself is easy to follow. Feature Propagation has a relatively small accuracy drop with a very high rate of missing features. Additionally, Feature Propagation is highly efficient and scales to extremely large graphs. The authors also theoretically consolidate the proposed method by proving that the feature propagation leads to a diffusion-type differential equation on the graph.

I have the following two suggestions.
1. In your main experiments, you use the commonly-used GCN model as the backbone encoder of FP. Can you provide experimental experiments based on more GNN backbone models, such as GAT, and GIN? It would be interesting to see how the backbone model affects the effectiveness of FP.
2. You conduct experiments with missing features. It naturally spurs a question: how FP performs with adversarial attacks on node features? Maybe this point is not the main problem you study in this paper, I think it would be interesting to leave it as future work.

---

### Official Review · Reviewer_eHaa · 2022-10-21

**Overall Score:** 8
**Confidence:** 3

**Review:**

#### Paper Summary
---
The paper proposes feature propagation (FP), a very simple approach that can be used as a preprocessing step for GNNs to recover missing node features based on both the known features and the graph structure. The authors build a theoretical connection between FP and Dirichlet energy minimization. The proposed method works well in practice even with as high as 99% missing features.

#### Strengths
---
- The paper is well written, and the problem is well-motivated.
- The proposed approach is very simple but has a nice theoretical foundation.
- Experimental evaluation is rigorous, and the method works well even at high rates of missing features.

#### Weaknesses
---
- I don't have any concerns.

#### Comments to Authors
---
I really enjoyed reading the paper. I vote for accepting this paper as I don't see any compelling reasons to reject it. Below are my minor comments:

1. As far as I know, apart from the two OGB datasets, the other five datasets used for evaluation come with sparse node features, among which four of them has binary features. How does the feature type affect the performance of the method? I think it would have been interesting to evaluate the effectiveness of the method under different types of node features (e.g., dense vs sparse, binary vs non-binary, categorical vs continuous, etc).

2. Looking at the results for 99% missing node features, we see that FP has only marginal improvement over a very simpler method like LP, which does not even need a GNN. I think the authors could have mentioned this fact that at extreme rates of missing features, sometimes it's better to switch to even simpler methods like LP, e.g., when the problem scale is very large, to reduce computational costs.

---

### Official Review · Reviewer_d76t · 2022-10-21

**Overall Score:** 6
**Confidence:** 4

**Review:**

Summary:
The paper proposes a method to reconstruct missing node features on the graph for downstream graph machine learning tasks. The solution is based on the minimization of the Dirichlet energy on the graph depending on the smoothness assumption of the features on the relational structure, a.k.a. homophily.
It heavily relies on the well-known label propagation method by Zhu & Ghahramani, and graph regularization studied by Zhou & Scholkopf.
Through discretization of the solution, the authors present an iterative algorithm called Feature Propagation (FP), which permits scalability to large graphs.
A composite pipeline is proposed for the node classification task (FP+GNN): the reconstructed node features are fed to a GNN classifier.
This pipeline is compared to two recent GNN frameworks and to the label propagation algorithm, which seems to show attaining state-of-the-art results on node classification task with missing node features.

Pros:
1. Computational effectiveness: scalability to large graphs
2. Effective in very high rates of missing features
3. The study emphasizes the problem of missing continuous, vectorial node features. There are very few graph machine learning studies addressing this. Previous approaches on semi-supervised learning on graphs were mainly limited to missing labeled nodes—only categorical features.

Cons/Concerns/Possible improvements:
1. The solution at the convergence is shown not depending on the initial values of the unknown features, yet initialization of them may affect the number of iterations leading to the convergence and also the accuracy of the solution. This seems to be alleviated by reseting the known features through the iterations of FP. Could that be also handled by preventing the propagation of randomly initialized node features onto other nodes?
2. The method is claimed to handle missing node features in graph machine learning tasks such as link prediction, graph classification, yet the experiments are limited to the node classification task only. Further analysis on the other tasks would be appreciated.
3. A table showing the statistics of the features of each dataset would be helpful to assess the feature imputation performance.
4. Evaluation of the feature imputation performance (without the downstream task) is also necessary to better asses the proposed FP algorithm.

Overlall, I vote for accept. Although the proposed feature propagation algorithm has limited novelty, it may stand as a step stone for further studies addressing learning graph representation with incomplete features, which is underadressed by the current GNN literature.

---

### Official Review · Reviewer_5ozq · 2022-10-24

**Overall Score:** 6
**Confidence:** 4

**Review:**

This paper proposes a general approach termed FP for handling missing features in graph machine learning applications that is based on minimization of the Dirichlet energy and leads to a diffusion-type differential equation on the graph. Feature Propagation is a simple, fast, scalable algorithm, and it can withstand surprisingly high rates of missing features.

Paper Strengths:

 (1) The paper gives the detailed derivation of the formula and the credible proof process of the theorem. It can be seen that the author has a strong theoretical foundation.

(2) The experiments in the article are sufficient, and the experiments on 7 datasets effectively verify the effectiveness of the proposed feature propagation algorithm, its speed and its advantages of being suitable for large-scale data.

Paper Weaknesses:

(1) The formulas given on line 136 in the main text and in line 532 in Appendix A.2 appear to be wrong because their results do not agree with the closed-form solution found in line 109 in the main text;

(2) Figure 4 is blurry, the details and numbers in the figure are not clear enough.

My Questions:

(1) The author mentions that the Feature Propagation algorithm is suitable for large-scale graph and runs very fast, but the complexity analysis of FP algorithm is not given in this article, and Eq. 2 does not seem to be of linear complexity. Please give an explanation;

(2) Why are the final convergence values for missing features from the formula in line 136 in the main text inconsistent with the closed-form solution in line 109 ?

 (3) From the experimental results given in Table 2, the Label Propagation algorithm can also be applied to large-scale graph-type data such as OGBN-products, and the effect achieved on downstream tasks is almost the same as the FP algorithm proposed in this paper. So what are the advantages of the proposed Feature Propagation algorithm compared to the Label Propagation approach ?

---

### Meta-Review · Area_Chair_adBD · 2022-11-16

**Confidence:** 5
**Recommendation:** Accept

**Meta Review:**

This paper proposes a new approach for handling missing features in graph machine learning applications that is based on minimization of the Dirichlet energy and leads to a diffusion-type differential equation on the graph. The proposed approach is simple and effective and also has a strong theoretical foundation. Experimental results show the effectiveness of the proposed method. In general, the paper is well written and easy to follow. The reviewers also mentioned some minor issues. The authors are encouraged to revise the paper based on the rebuttal and comments by reviewers. For example, the authors can provide experimental experiments based on more GNN backbone models, such as GAT, and GIN.

---

### Decision · Program_Chairs · 2022-11-22

Accept (Poster)